# PROCEEDINGS A

applied mathematics, mathematical modelling

model selection, hybrid systems, information criteria, sparse regression, nonlinear dynamics, data-driven discovery

**Author for correspondence:**
N. M. Mangan
e-mail: niallmm@gmail.com

# Model selection for hybrid dynamical systems via sparse regression

N. M. Mangan[1], T. Askham[2], S. L. Brunton[4], J. N. Kutz[2] and J. L. Proctor[3]

[1]Department of Engineering Sciences and Applied Mathematics, Northwestern University, Evanston, IL 60208, USA
[2]Department of Applied Mathematics, and [3]Department of Mechanical Engineering, University of Washington, Seattle, WA 98195, USA
[4]Institute for Disease Modeling, Bellevue, WA 98005, USA

NMM, 0000-0002-3491-8341

Hybrid systems are traditionally difficult to identify and analyse using classical dynamical systems theory. Moreover, recently developed model identification methodologies largely focus on identifying a single set of governing equations solely from measurement data. In this article, we develop a new methodology, Hybrid-Sparse Identification of Nonlinear Dynamics, which identifies separate nonlinear dynamical regimes, employs information theory to manage uncertainty and characterizes switching behaviour. Specifically, we use the nonlinear geometry of data collected from a complex system to construct a set of coordinates based on measurement data and augmented variables. Clustering the data in these measurement-based coordinates enables the identification of nonlinear hybrid systems. This methodology broadly empowers nonlinear system identification without constraining the data locally in time and has direct connections to hybrid systems theory. We demonstrate the success of this method on numerical examples including a mass–spring hopping model and an infectious disease model. Characterizing complex systems that switch between dynamic behaviours is integral to overcoming modern challenges such as eradication of infectious diseases, the design of efficient legged robots and the protection of cyber infrastructures.

# 1. Introduction

The high-fidelity characterization of complex systems is of paramount importance to manage modern infrastructure and improve lives around the world. However, when a system exhibits nonlinear behaviour and switches between dynamical regimes, as is the case for many large-scale engineered and human systems, model identification is a significant challenge. These *hybrid systems* are found in a diverse set of applications including epidemiology [1], legged locomotion [2], cascading failures on the electrical grid [3] and security for cyber infrastructure [4]. Typically, model selection procedures rely on physical principles and expert intuition to postulate a small set of candidate models; information theoretic approaches evaluate the goodness of fit to data among these models and penalizing over-fitting [5–9].

More candidate models can be considered using advanced data-driven methodologies such as support vector machines [10,11], Bayesian variable selection [12,13], genetic algorithms [14–16] and information theoretic techniques [17]. In 2016, Wang *et al.* [18] provided a nice review of data-driven identification of complex systems. Our contribution, *sparse identification of nonlinear dynamics* (SINDy) [19], sparsely selects models from a combinatorially large library of possible nonlinear dynamical systems, decreases the computational costs of model fitting and evaluations [20] and generalizes to a wide variety of physical phenomena [21,22]. However, neither standard nor advanced model selection procedures are formulated to identify hybrid systems. In this article, we describe a new method, called Hybrid-Sparse Identification of Nonlinear Dynamics (Hybrid-SINDy), which identifies hybrid dynamical systems, characterizes switching behaviours and uses information theory to manage model selection uncertainty.

Predecessors of current data-driven model-selection techniques, called *system identification*, were developed by the controls community to discover linear dynamical systems directly from data [23]. They made substantial advances in the model identification and control of aerospace structures [24,25], and these techniques evolved into a standard set of engineering control tools [26]. One method to improve the prediction of linear input–output models was to augment present measurements with past measurements, i.e. delay embeddings [24]. Delay embeddings and their connections to Takens' embedding theorem have enabled equation-free techniques that distinguish chaotic attractors from measurement error in time-series [27], contribute to nonlinear forecasting [28,29] and identify causal relationships among subsystems solely from time-series data [30].

Augmenting measurements with nonlinear transformations has also enabled identification of nonlinear dynamical systems from data. As early as 1987, nonlinear feature augmentation was used to construct equations and characterize the dynamical system [31], with extensions to control in 1991 [32]. Later, ordinary differential equations were formulated into a dictionary of observables, through linear discretization of the derivatives, and convergence and error of discretization schemes of varying order were studied [33]. Developed more recently, dynamic mode decomposition [34–36] has been connected to nonlinear dynamical systems via the Koopman operator [35,37,38], and extended to control [39]. More sophisticated data transformations, originating in the harmonic analysis community, are also being used for identifying nonlinear dynamical systems [40,41]. Similarly, SINDy exploits these nonlinear transformations by building a library of nonlinear dynamic terms constructed using data. This library is systematically refined to find a parsimonious dynamical model that represents the data with as few nonlinear terms as possible [19].

Methods like SINDy are not currently designed for hybrid systems because they assume that all measurement data in time are collected from a dynamical system with a consistent set of equations. In hybrid systems, the equations may change suddenly in time and one would like to identify the underlying equations without knowledge of the switching points. One approach is to construct the models locally in time by restricting the input data to a short time window. Statistical models, such as the auto-regressive moving average (ARMA) and its nonlinear counterpart (NARMA), constrain the time series to windows of data near the current time [42]. This technique

has been extended to analyse non-autonomous dynamical systems, including hybrid systems, with Koopman operator theory [43].

Other approaches for nonlinear systems include partitioning the time- or spatial-domain and constructing local models using Galerkin's method for model reduction, proper orthogonal decomposition or dynamic mode decomposition to construct reduced-order models [44–47]. Cluster-based approaches have also been used to build probabilistic, reduced-order models of complex fluid flow [48]. A recent method used a global optimum search to optimize the number of clusters and generated local reduced-order models using proper orthogonal decomposition for computational speed up of simulations of hydraulic fracturing [49]. Subsequent work incorporated dynamic mode decomposition with control to design an approximate model with feedback control on local temporal clusters [50], and then extended to handle spatial heterogeneity using an ensemble Kalman filter [51]. Reduced-order models for simulating fracture propagation have also been developed using SINDy [52]. Alternatively, recent methods for recurring switching between dynamical systems use a Bayesian framework to infer how the state of the system, modelled as linear partitions, depends on multiple previous time steps [53]. This method enables reconstruction of state space in terms of linear generated states and provides location-dependent behavioural states.

While restricting data locally in time may avoid erroneous model selection at the switching point, this method creates a new problem: there may not be enough data within a single window for data-driven model selection to robustly select and validate nonlinear models. For some sparse regression problems, only a small amount of data is required to accurately recover the signal. Schaeffer *et al.* recently demonstrated dynamical system recovery in [54,55], using short bursts from random initial conditions and a Legendre polynomial basis. They take advantage of recovery guarantees for basis pursuit when library features are relatively uncorrelated and measurement noise is limited [56]. However, the recovery guarantees fail for short time series from the mass–spring hopper system analysed in this work [57]. Furthermore, unlike the systems analysed in [54,55], one cannot collect randomly sampled initial conditions from a hybrid dynamical system because the data would bridge multiple dynamic regimes. Another standard sparsity promoting technique, LASSO, has been recently shown to make mistakes early in the sparse recovery pathway [58], whereas the least squares with thresholding procedure advocated here converges locally to the solution of a non-convex, $\ell_0$-penalized regression problem and such non-convex methods have been observed to outperform convex variants in sparse variable selection [59,60]. Even when low-data-limit recovery guarantees exist for an appropriate sparsity promoting method, more data are necessary to validate the recovered system. Most sparse-regression methods have a tuning parameter which generates a collection of models of varying sparsity, therefore a significant amount of local validation data is required to differentiate between these models.

For nonlinear-model selection and validation to work in hybrid-systems, one needs a method to gather sufficient data from a consistent underlying model. Simplex-projection, which is used in cross convergent mapping, employs delay embeddings to find geometrically similar data for prediction [27]. Recently, Yair *et al.* showed that data from dynamically similar systems could be grouped together in a label-free way by measuring geometric closeness in the data using a kernel method [61]. Here, we show that nonlinear model selection can succeed for hybrid dynamical systems when the data are examined within a pre-selected coordinate system that takes advantage of the intrinsic geometry of the data.

We present a generalization of SINDy, called Hybrid-SINDy, that allows for the identification of nonlinear hybrid dynamical systems. We use modern machine-learning methodologies to identify clusters within the measurement data augmented with features extracted from the measurements. Applying SINDy to these clusters generates a library of candidate nonlinear models. We demonstrate that this model library contains the different dynamical regimes of a hybrid system and use out-of-sample validation with information theory to identify switching behaviour. We perform an analysis of the effects of noise and cluster size on model recovery. Hybrid-SINDy is applied to two realistic applications including legged locomotion

and epidemiology. These examples span two fundamental types of hybrid systems: time- and state-dependent switching behaviours.

## 2. Background

### (a) Hybrid systems

Hybrid systems are ubiquitous in biological, physical and engineering systems [1–4]. Here, we consider hybrid models in which continuous-time vector fields describing the temporal evolution of the system state *change* at discrete times, also called events. Specifically, we choose a framework and definition for hybrid systems that is amenable to numerical simulations [62] and has been extensively adapted and used for the study of models [2]. Note that these models are more complicated to define, numerically simulate and analyse than classical dynamical systems with smooth vector fields [62,63]. Despite these challenges, solutions of these hybrid models have an intuitive interpretation: the solution is composed of piecewise continuous trajectories evolving according to vector fields that may change discontinuously at events.

Consider the state space of a hybrid system as a union

$$V = \bigcup_{\alpha \in I} V_\alpha, \tag{2.1}$$

where $V_\alpha$ is a connected open set in $\mathbb{R}^n$ called a chart and $I$ is a finite index. Describing the state of the system requires an index $\alpha$ and a point in $V_\alpha$, which we denote as $\mathbf{x}^\alpha$. We assume that the state within each patch evolves according to the classic description of a dynamical system $\dot{\mathbf{x}}^\alpha(t) = \mathbf{f}^\alpha(\mathbf{x}^\alpha(t))$, where $\mathbf{f}^\alpha(\mathbf{x}^\alpha)$ represents the governing equations of the system for chart $V_\alpha$. Transition maps $T^\alpha$ apply a change of states to boundary points within the chart; see [2] for a more rigorous definition of $T^\alpha$. In this work, we consider hybrid systems where the transition between charts links the final state of the system on one chart $\mathbf{x}_{\alpha_i}$ to the initial condition on another $\mathbf{x}_{\alpha_j}$ where both $\mathbf{x}_{\alpha_i}, \mathbf{x}_{\alpha_j} \in \mathbb{R}^n$. Constructing the global evolution of the system *across* patches requires concatenating a set of smooth trajectories separated by a series of discrete events in time $\tau_1, \tau_2, \ldots, \tau_o$. These discrete events can be triggered by either the state of the system $\tau_i(\mathbf{x})$ or external events in time $\tau_i(t)$. In this article, we analyse hybrid systems representing both state- and time-dependent events. For a broader and more in-depth discussion on hybrid systems, we refer the reader to [2,62,63].

### (b) Sparse identification of nonlinear dynamics

SINDy combines sparsity-promoting regression and nonlinear function libraries to identify a nonlinear, dynamical system from time-series data [19]. We consider dynamical systems of the form

$$\frac{\mathrm{d}}{\mathrm{d}t}\mathbf{x}(t) = \sum_{l=1}^{\zeta} \xi_l \mathbf{f}_l(\mathbf{x}(t)), \tag{2.2}$$

where $\mathbf{x}(t) \in \mathbb{R}^n$ is a vector denoting the state of the system at time $t$ and the sum of functions $\sum_{l=1}^{\zeta} \xi_l \mathbf{f}_l$ describes how the state evolves in time. Importantly, we assume that $\zeta$ is small, indicating the dynamics can be represented by a parsimonious set of basis functions. To identify these unknown functions from known measurements $\mathbf{x}(t)$, we first construct a comprehensive library of candidate functions $\boldsymbol{\Theta}(\mathbf{x}) = [\mathbf{f}_1(\mathbf{x}) \, \mathbf{f}_2(\mathbf{x}) \, \ldots \, \mathbf{f}_p(\mathbf{x})]$. We assume that the functions in (2.2) are a subset of $\boldsymbol{\Theta}(\mathbf{x})$. The measurements of the state variables are collected into a data matrix $\mathbf{X} \in \mathbb{R}^{(m \times n)}$, where each row is a measurement of the state vector $\mathbf{x}^{\mathrm{T}}(t_i)$ for $i \in [1, m]$. The function library is then evaluated for all measurements $\boldsymbol{\Theta}(\mathbf{X}) \in \mathbb{R}^{(m \times p)}$. The corresponding derivative time-series data, $\dot{\mathbf{X}} \in \mathbb{R}^{(m \times n)}$, are either directly measured or numerically calculated from $\mathbf{X}$.

To identify (2.2) from the data pair $(\boldsymbol{\Theta}(\mathbf{X}), \dot{\mathbf{X}})$, we solve

$$\dot{\mathbf{X}} = \boldsymbol{\Theta}(\mathbf{X})\boldsymbol{\Xi}, \tag{2.3}$$

for the unknown coefficients $\boldsymbol{\Xi} \in \mathbb{R}^{(p \times n)}$ and enforce a penalty on the number of non-zero elements in $\boldsymbol{\Xi}$. Note that the $i$th column of $\boldsymbol{\Xi}$ determines the governing equation for the $i$th state variable. We expect each coefficient vector in $\boldsymbol{\Xi}$ to be sparse, such that only a small number of elements are non-zero. We can find a sparse-coefficient vector using the Lagrangian minimization problem

$$\min_{\boldsymbol{\Xi}} \frac{1}{2}\|\dot{\mathbf{X}} - \boldsymbol{\Theta}(\mathbf{X})\boldsymbol{\Xi}\|_2^2 + \hat{\lambda}R(\boldsymbol{\Xi}). \tag{2.4}$$

Here, $R(\boldsymbol{\Xi})$ is a regularizing, sparse-penalty function in terms of the coefficients, and $\hat{\lambda}$ is a free parameter that controls the magnitude of the sparsity penalty. Two commonly used formulations include the *LASSO* with an $l_1$ penalty $R(\boldsymbol{\Xi}) = \|\boldsymbol{\Xi}\|_1$ and the elastic-net with an $l_1$ and $l_2$ penalty $R(\boldsymbol{\Xi}) = \gamma\|\boldsymbol{\Xi}\|_1 + \frac{1}{2}(1-\gamma)\|\boldsymbol{\Xi}\|_2^2$ which includes a second free parameter $\gamma$ [64]. Less common, but perhaps more natural, is the choice $R(\boldsymbol{\Xi}) = \|\boldsymbol{\Xi}\|_0$, where the $\ell_0$ penalty is given by the number of non-zero entries in $\boldsymbol{\Xi}$. In this article, we use sequential least squares with hard thresholding to solve (2.4) with the $\ell_0$-type penalty, where any coefficients with values less than a threshold $\lambda$ are set to zero in each iteration [19].

Several innovations have followed the original formulation of SINDy [19]: the framework has been generalized to study partial differential equations [22,65] and systems with rational functional forms [21]; the impact of highly corrupted data has been analysed [66]; the robustness of the algorithm to noise has been improved using integral and weak formulations [67,68]; and the theory has been generalized to non-autonomous dynamical system with time-varying coefficients using group sparsity norms [69,70]. Additional connections with information criteria [20], and extensions to incorporate known constraints, for example, to enforce energy conservation in fluid flow models [71], have also been explored. The connection with the Akaike information criteria (AIC) is essential for this work, as it allows automated evaluation of SINDy-generated models.

## (c) Model selection using Akaike information criteria

Information criteria provide a principled methodology to select between candidate models for systems without a well-known set of governing equations derived from first principles. Historically, experts heuristically constructed a small number, $\mathcal{O}(10)$, of models based on their knowledge or intuition [72–77]. The number of candidate models is limited due to the computational complexity required in fitting each model, validating on out-of-sample data and comparing across models. New methods, including SINDy, identify data-supported models from a much larger space of candidates without constructing and simulating every model [14,19,78,79]. The fundamental goal of model selection is to find a parsimonious model, which minimizes error without adding unnecessary complexity through additional free parameters.

In 1951, Kullback and Leibler (K–L) proposed a method for quantifying information loss or 'divergence' between reality and model predictions [80]. Akaike subsequently calculated the relative information loss between models, by connecting K–L divergence theory with the likelihood theory from statistics. He discovered a deceptively simple estimator for computing the relative K–L divergence in terms of the maximized log-likelihood function for the data given a model, $L(\mathbf{x}, \hat{\mu})$, and the number of free parameters, $k$ [5,6]. This relationship is now called AIC:

$$\text{AIC} = 2k - 2\ln(L(\mathbf{x}, \hat{\mu})), \tag{2.5}$$

where the observations are $\mathbf{x}$, and $\hat{\mu}$ is the best-fit parameter values for the model given the data. The maximized log-likelihood calculation is closely related to the standard ordinary least squares when the error is assumed to be independently, identically, and normally distributed (IIND). In this special case, $\text{AIC} = \rho\ln(\text{RSS}/\rho) + 2k$, where RSS is the residual sum of the squares and $\rho$ is the number of observations. The RSS is expressed as $\text{RSS} = \sum_{i=1}^{\rho}(y_i - g(x_i; \mu))^2$ where $y_i$ are the

observed outcomes, $x_i$ are the observed independent variables, and $g$ is the candidate model [72]. Note that the RSS and the log-likelihood are closely connected.

In practice, the AIC requires a correction for finite sample sizes given by

$$\text{AIC}_c = \text{AIC} + \frac{2(k+1)(k+2)}{(\rho - k - 2)}. \tag{2.6}$$

AIC and $\text{AIC}_c$ contain arbitrary constants that will depend on the sample size. These constants cancel out when the minimum $\text{AIC}_c$ across models is subtracted from the $\text{AIC}_c$ for each candidate model $j$, producing an interpretable model selection indicator called relative $\text{AIC}_c$, described by $\Delta\,\text{AIC}_c^j = \text{AIC}_c^j - \text{AIC}_c^{\min}$. The model with the most support will have a score of zero; $\Delta\,\text{AIC}_c$ values allows us to rank the relative support of the other models. Anderson and Burnham in their seminal work [72] prescribe a general rule of thumb when comparing relative support among models: models with $\Delta\,\text{AIC}_c < 2$ have substantial support, $4 < \Delta\,\text{AIC}_c < 7$ have some support, and $\Delta\,\text{AIC}_c > 10$ have little support. These thresholds directly correspond to a standard $p$-value interpretation; we refer the reader to [72] for more details. In this article, we use $\Delta\,\text{AIC}_c = 3$ as a slightly larger threshold for support in this study. Following the development of AIC, many other information criteria have been developed including Bayesian information criterion (BIC) [81], cross-validation (CV) [82], deviance information criterion (DIC) [83] and minimum description length (MDL) [84]. However, AIC remains a well known and ubiquitous tool; in this article, we use relative $\text{AIC}_c$ with correction for low data-sampling [20].

## 3. Hybrid-SINDy

Hybrid-SINDy is a procedure for augmenting the measurements, clustering the measurement and augmented variables and selecting a model using SINDy for each cluster. We describe how to validate these models and identify switching between models. An overview of the hybrid-SINDy method is provided in figure 1 and algorithm 1.

### (a) Collect time-series data from system

Discrete measurements of a dynamical system are collected and denoted by $\mathbf{x}(t_i) \in \mathbb{R}^n$; see figure 1$b$ for a time-series plot of the hopping robot illustrated in figure 1$a$. The measurement data are arranged into the matrix $\mathbf{X} = [\mathbf{x}(t_1)\,\mathbf{x}(t_2)\,\ldots\,\mathbf{x}(t_b)]^T \in \mathbb{R}^{(b \times n)}$, where superscript 'T' is the matrix transpose. The time series may include trajectories from multiple initial conditions concatenated together. The SINDy model is trained with a subset of the data $\mathbf{X}_T \in \mathbb{R}^{(m \times n)}$, where $m$ is the number of training samples. The corresponding data matrices for validation are denoted $\mathbf{X}_V \in \mathbb{R}^{v \times n}$, where $v$ is the number validation samples, and $b = m + v$.

### (b) Clustering in measurement-based coordinates

Applications may require augmentation with variables such as the derivative, nonlinear transformations [40,41], or time-delay coordinates [24,28]. In this article, we augment the state measurements $\mathbf{x}(t_i)$ with the time derivative of the measurements. The time derivative matrix is constructed similar to the measurement matrix $\dot{\mathbf{X}}_T = [\dot{\mathbf{x}}(t_1)\,\dot{\mathbf{x}}(t_2)\,\ldots\,\dot{\mathbf{x}}(t_m)]^T \in \mathbb{R}^{(m \times n)}$. The matrices $\dot{\mathbf{X}}_T$ and $\dot{\mathbf{X}}_V$ are either directly measured or calculated from $\mathbf{X}_T$ and $\mathbf{X}_V$, respectively. If all state variables are accessible, such as in a numerical simulation, these data-driven coordinates directly correspond to the phase space of a dynamical system. Note that this coordinate system does not explicitly incorporate temporal information. Figure 1$c$ illustrates the coordinates $(x, \dot{x})$ for the hopping robot. A subset of $[\mathbf{X}_T\,\dot{\mathbf{X}}_T]$ can also be used as measurement-based coordinates. The set of indices $D$ are the measurements (columns), which are included in the analysis denoted by $\mathbf{Y}_T$ and $\mathbf{Y}_V$.

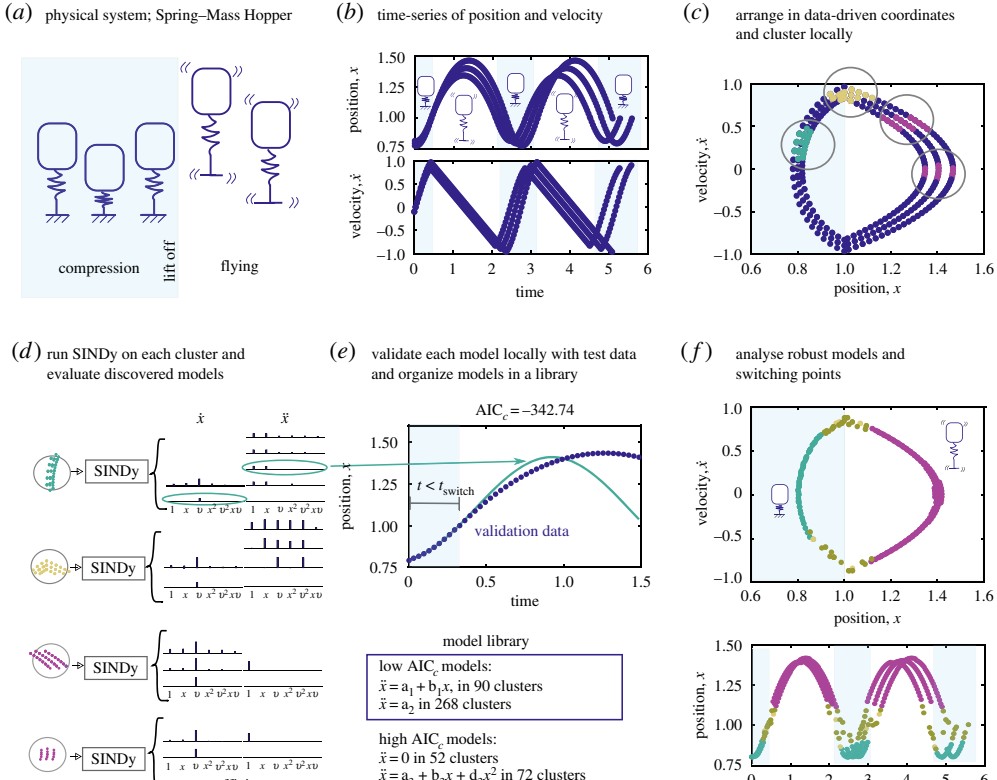

**Figure 1.** Overview of the hybrid-SINDy method, demonstrated using the Spring–Mass Hopper system. (*a*) The two dynamic regimes of spring compression (blue) and flying (white). Time series for the position and velocity of the system sample both regimes (*b*). Clustering the data in data-driven coordinates allows separation of the regimes, except at transition points near $x = 1$ (*c*). Performing sparse model selection on each cluster produces a number of possible models per cluster (*d*). (*e*) Validating each model within the cluster to form a model library containing low $AIC_c$ models across all clusters. In (*f*), we plot the location of the four most frequent models across clusters. These models correctly identify the compression, flying and transition points. (Online version in colour.)

We then identify clusters of samples in the training and validation sets. For each sample (row) in $\mathbf{Y}_T$, we use the nearest neighbour algorithm *knnsearch* in MATLAB to find a cluster of $K$ similar measurements in $\mathbf{Y}_T$. The training-set clusters, which are row indices of $\mathbf{Y}_T$ denoted $\mathbf{C}_T^i \in \mathbb{R}^K$, are found for each time point $t_i \in [t_1, t_2, \ldots, t_m]$. The centroid of each cluster is computed within the training set $\mathbf{Y}_T(\mathbf{C}_T^i)$. We then identify $K$ measurements from $\mathbf{Y}_V$ near the *training* centroid clusters. Note that these clusters in the validation data, $\mathbf{C}_V^i \in \mathbb{R}^K$, are essential to testing the out-of-sample prediction of Hybrid-SINDy. Figure 2*a*,*b* illustrates the validation set in measurement-based coordinates, with the centroids of three training clusters as black dots and the corresponding validation clusters in teal, gold and purple dots.

By finding the corresponding validation clusters, we ensure that the out-of-sample data for validating the model have the same local, nonlinear characteristics of the training data. To assess the performance of the models, we also need to identify a validation time series from $\mathbf{Y}_V$. Starting with each data point in a validation cluster $\mathbf{C}_V^i$, we collect $q$ measurements from $\mathbf{Y}_V$ that are temporally sequential, where $q \ll m$. These subsets of validation time series, $\mathbf{Z}_V^i \in \mathbb{R}^{q \times n}$, are defined for each data point and each cluster. The validation time series helps characterize the out-of-sample performance of the model fit.

---

**Algorithm 1**. Hybrid-SINDy.

---

**Input:** The measurement data $\mathbf{X} \in \mathbb{R}^{b \times n}$, the set of measurement variables $D \in \mathbb{R}^{d \times 1}$ for clustering, the length of validation time series $q$, the number of data points in the training set $m$, the number of data points in the validation set $v$, the sparsification values $\boldsymbol{\lambda} \in \mathbb{R}^r$, the number of library terms $p$, and the number of samples in each cluster $K$.

1: **procedure** HYBRID-SINDY($\mathbf{X}, D, s, m, v, c, K$)
2: $\mathbf{X}_T \in \mathbb{R}^{m \times n}, \mathbf{X}_V \in \mathbb{R}^{v \times n} \leftarrow$ testTrainSeparation($\mathbf{X}, m, v$) ▷ Construct training/validation
3: $\dot{\mathbf{X}}_T \in \mathbb{R}^{m \times n}, \dot{\mathbf{X}}_V \in \mathbb{R}^{v \times n} \leftarrow$ derivative($\mathbf{X}_T, \mathbf{X}_V$) ▷ Compute derivative matrix
4: $\mathbf{Y}_T \in \mathbb{R}^{m \times d} \leftarrow$ variables($\mathbf{X}_T, \dot{\mathbf{X}}_T, D$) ▷ Construct Augmented Measurements
5: $\mathbf{Y}_V \in \mathbb{R}^{m \times d} \leftarrow$ variables($\mathbf{X}_V, \dot{\mathbf{X}}_V, D$) ▷ Construct Augmented Measurements
6: **for** $i \in \{1, 2, \ldots m\}$ **do** ▷ For each sample in the training set $t_i$, compute:
7: $\mathbf{C}_T^i \in \mathbb{R}^K \leftarrow$ cluster($\mathbf{Y}_T, \mathbf{Y}_T(i, :), K$) ▷ Cluster K samples from $\mathbf{Y}_T$ for each $\mathbf{Y}_T(i, :)$
8: $\mathbf{C}_V^i \in \mathbb{R}^K \leftarrow$ cluster($\mathbf{Y}_V$, centroid($\mathbf{Y}_T(\mathbf{C}_T^i, :)$), $K$) ▷ Cluster K samples of $\mathbf{Y}_V$ for $\mathbf{C}_T^i$
9: $\boldsymbol{\Theta}^i \in \mathbb{R}^{m \times p} \leftarrow$ library($\mathbf{X}_T(\mathbf{C}_T^i, :)$) ▷ Generate library that contains $p$ features
10: **for** $j \in \{1, 2, \cdots, r\}$ **do** ▷ Search over sparsification parameter $\boldsymbol{\lambda}$.
11: Model($j$) $\leftarrow$ SINDy($\dot{\mathbf{X}}_T(\mathbf{C}_T^i, :), \boldsymbol{\Theta}^i, \lambda(j)$ ) ▷ Identify sparse features & model.
12: **for** $s \in \{1, 2, \ldots, K\}$ **do** ▷ Calculate error for each point in cluster
13: $\mathbf{Z} \in \mathbb{R}^{q \times n} \leftarrow$ simulate (Model($j$), $\mathbf{X}_V(\mathbf{C}_V^i(s), :), q$) ▷ Simulate model
14: $\mathbf{Z_V} \in \mathbb{R}^{q \times n} \leftarrow$ find($\mathbf{X}_V, \mathbf{X}_V(\mathbf{C}_V^i(s), :), q$) ▷ Find validation time series
15: $t_s \leftarrow$ detect switching($\mathbf{Z}, \mathbf{Z}_V$) ▷ Find switching time
16: **for** $l \in \{1, 2, \ldots, n\}$ **do** ▷ Calculate error
17: $E_{variable}(l) \leftarrow \frac{1}{t_s} \sum_{a=1}^{t_s} (\mathbf{Z}(a, l) - \mathbf{Z}_V(a, l))^2$ ▷ Avg. over time
18: **end for**
19: $E_{avg}(s) \leftarrow \frac{1}{n} \sum_{l=1}^n E_{variable}(l)$ ▷ Avg. over measurements
20: **end for**
21: k $\leftarrow$ numberOfFreeParameters(Model($j$))
22: AIC$_c$($j$) $\leftarrow$ ComputeAIC$_c$($E_{avg}$, k, K)
23: **end for**
24: $\Delta$ AIC$_c$ $\leftarrow$ sort(AIC$_c$ $-$ minimum(AIC$_c$)) ▷ Rank models by relative AIC$_c$ scores.
25: $\boldsymbol{\Pi} \leftarrow \mathbf{I}$(Model($\Delta$ AIC$_c < 3$)) ▷ Store models with support in library
26: **end for**
27: ind $\leftarrow$ sort(frequency($\boldsymbol{\Pi}$)) ▷ Sort models by frequency across clusters.
28: **return** $\boldsymbol{\Pi}(ind)$ ▷ return the most frequent models.
29: **end procedure**

---

## (c) SINDy for clustered data

We perform SINDy for each training cluster $\mathbf{C}_T^i$, using an alternating least squares and hard thresholding described in [19] and §2*b*. For each cluster, we search over the sparsification parameter, $\lambda(j) \in \{\lambda_1, \lambda_1, \ldots \lambda_r\}$, generating a set of candidate models for each cluster; see figure 1*d* for an illustration. In practice, the number of models per cluster is generally less than $r$ since multiple values of $\lambda$ can produce the same model. In this article, the library, $\boldsymbol{\Theta}(\mathbf{X})$, includes polynomial functions of increasing order (i.e. $x, x^2, x^3, \ldots$), similar to the examples in [19]. However, the SINDy library can be constructed with other functional forms that reflect intuition about the underlying process and measurement data.

## (d) Model validation and library construction

Validation involves producing simulations from candidate models and comparing to the validation data. Using the validation cluster as a set of $K$ initial conditions $\mathbf{C}_V^i$, we simulate each candidate model $j$ in cluster $i$ for $q$ time steps producing time series $\mathbf{Z} \in \mathbb{R}^{q \times n}$. We compare these simulations against the validation time series $\mathbf{Z}_V$ and calculate an out-of-sample AIC$_c$ score. An example illustration comparing $\mathbf{Z}_V$ and $\mathbf{Z}$ for a single cluster is shown in figure 2*a*.

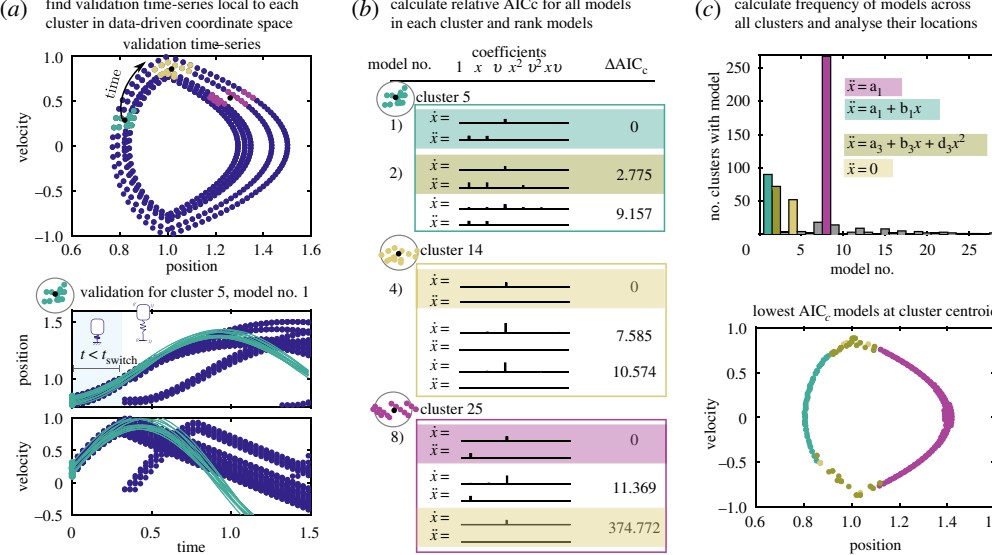

**Figure 2.** Steps for local validation and selection of models. For each cluster from the training set, we identify validation time-series points that are local to the training cluster centroid (black dots, (*a*)). We simulate time series for each model in the cluster library, starting from each point in the validation cluster (teal, gold and purple dots) and calculate the error from the validation time series. Using this error we calculate a relative $AIC_c$ value and rank each model in the cluster (*b*). We collect the models with significant support into a library, keeping track of their frequency across clusters. The highest frequency models across clusters are shown in (*c*). Note that the colours associated with each model in (*c*) are consistent across panels. (Online version in colour.)

In order to calculate the error between the simulation and validation, we must first account for the possibility of the dynamics switching before the end of the $q$ validation time steps. We use the function *findchangepoints* in Matlab [85] to detect a change in the mean of the absolute error between the simulated and validation time series. The time index closest to this change is denoted $t_s$. Notably, this algorithm does not robustly find the time at which our time-series switch dynamical regimes. The algorithm tends to identify the transition prematurely, especially in oscillatory systems. We use $t_s$ as a *lower bound*, before which we can reasonably compare the simulated and validation data.

To assess a model's predictive performance within a cluster, we compare the simulated data $\mathbf{Z}$ and validation data $\mathbf{Z}_V$ restricted to time points before $t_s$. Specifically, we calculate the residual sum of square error for a candidate model by comparing the $K$ time series from the validation data to the model outputs, described by $E_{\mathrm{avg}}(s) = (1/n) \sum_{l=1}^{n} ((1/t_s) \sum_{a=1}^{t_s} (z_{a,l} - z_{a,l}^V)^2)$ for $s \in [1, K]$, where $z_{a,l}$ corresponds to the $a$ row and $l$ column of $\mathbf{Z}$ and similarly with $z_{a,l}^V$ to $\mathbf{Z}_V$. Thus, the vector $E_{\mathrm{avg}}$ contains the average error over time points and state variables for $K$ initial conditions of model $r$.

For each candidate model $r$, we calculate the $AIC_c$ from (2.6) using $AIC(r) = K \ln(E_{\mathrm{avg}}/K) + 2k$, the number of initial conditions in the validation set $K$, the average error for each initial condition, $E_{\mathrm{avg}}$, and the number of free parameters (or terms) in the selected model $k$ [5,6]. An equivalent procedure is found in [20]. Once we have $AIC_c$ scores for each model within the cluster, we calculate the relative $AIC_c$ scores and identify models within the cluster with significant support where the relative $AIC_c < 3$; see figure 2*b* for an illustration. These models are used to build the model library. Models with larger relative $AIC_c$ are discarded, illustrated in figure 1*e*. Note that multiple models can have significant support within a single cluster. We include each of these supported models in the library. The model library records the structure of highly supported models and how many times they appear across clusters.

Choosing the optimal number of data points $K$ for all clusters will be application specific. For too few data points per cluster, the out-of-sample error should be large due to model

misidentification. Increasing the *K* value should decrease the out-of-sample error and mitigate the impact of noise; choosing a specific *K* will require the practitioner to decide on an acceptable out-of-sample error profile. For large *K* more clusters will include a switching point, resulting in misidentification as the cluster will include data from multiple processes. This will appear as a rise in out-of-sample error and effectively decrease the resolution of switching point discovery.

## (e) Identification of high-frequency models and switching events

After building a library of strongly supported models, we analyse the frequency of model structures appearing across clusters, illustrated in figure 2*c*. The most frequent models and the location of their centroids provide insight into connected regions of measurement space with the same model (e.g. figure 1*f*). By examining the location and absolute $AIC_c$ scores of the models, we can identify regions of similar dynamic behaviour and characterize events corresponding to dynamic transitions.

# 4. Results: model selection

## (a) Mass–spring hopping model

In this subsection, we demonstrate the effectiveness of Hybrid-SINDy by identifying the dynamical regimes of a canonical hybrid dynamical system: the spring–mass hopper. The switching between the flight and compression stages of the hopper depends on the state of the system [2]. Figure 1*b* illustrates the flight and spring compression regimes and dynamic transitions. Note these distinct dynamical regimes are called charts, and liftoff and touchdown points are state-dependent events separating the dynamical regimes; see §2*a* for connections to hybrid dynamical system theory. The legged locomotion community has been focused on understanding hybrid models due to their unique dynamic stability properties [86], the insight into animal and insect locomotion [2,87], and guidance on the construction and control of legged robots [88–90].

A minimal model of the spring–mass hopper is given by the following:

$$m\ddot{x} = \begin{cases} -k(x - x_0) - mg, & x \leq x_0, \\ -mg, & x > x_0, \end{cases} \tag{4.1}$$

where *m* is the mass, *k* is the spring constant, and *g* is the gravity. The unstretched spring length $x_0$ defines the flight and compression stages, i.e. $x > x_0$ and $x \leq x_0$, respectively. For convenience, we non-dimensionalize (4.1) by scaling the height of the hopper by $y = x/x_0$, scaling time by $\tau = t\sqrt{(kx_0/m)}$ and forming the non-dimensional parameter $\kappa = kx_0/mg$. Thus, $\kappa$ represents the balance between the spring and gravity forces. Equation (4.1) becomes

$$\ddot{y} = \begin{cases} 1 - \kappa(y - 1), & y \leq 1, \\ -1, & y > 1. \end{cases} \tag{4.2}$$

For our simulations, we chose $\kappa = 10$. The switching point between compression and flying occurs at $y = 1$ in this non-dimensional formulation.

### (i) Generating input time series from the model

We generate time-series samples from (4.2) by selecting three initial conditions $(y_0, \dot{y}_0) \in \{(0.8, -0.1), (0.78, -0.1), (0.82, -0.1)\}$. We simulate the system for a duration of $t = [0\ 5]$ with sampling intervals of $\Delta\tau = 0.033$, producing 152 samples per initial condition. The resulting time series of the position and velocity, $\mathbf{y}(t_i)$ and $\dot{\mathbf{y}}(t_i)$, are used to construct the training-set matrices $\mathbf{X}_T = \mathbf{Y}_T = [\mathbf{y}(t_i)\ \dot{\mathbf{y}}(t_i)]$ where each row corresponds to sample. The position and velocity time series are plotted in figure 1*b*. Figure 1*c* illustrates the position–velocity

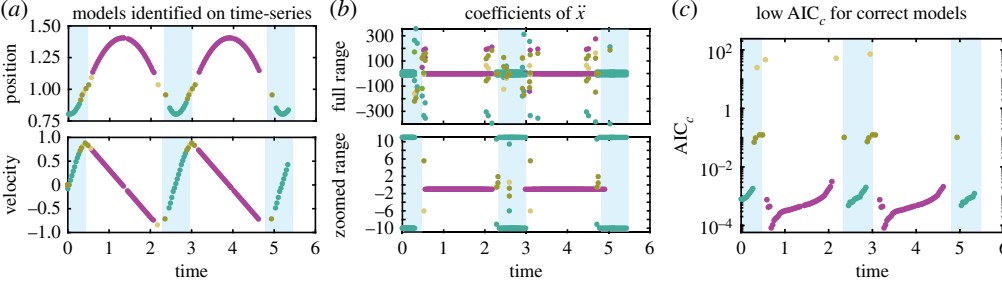

**Figure 3.** Hopping model discovery shown in time. Single time series of data and associated model coefficients and $AIC_c$ are plotted as a function of time with the correct models indicated by colour. Teal dots indicate the recovery of compressed spring model, purple dots indicate recovery of the flying model and yellow and gold dots indicate recovery of incorrect models. Both coefficients and absolute $AIC_c$ are plotted for the model with only the lowest $AIC_c$ value at each cluster. (Online version in colour.)

trajectories in phase-space. We also add Gaussian noise with mean zero and standard deviation $10^{-6}$ to the position and velocity time series in $\mathbf{Y}_T$. In this example, the derivatives $\dot{\mathbf{X}}_T = \dot{\mathbf{Y}}_T = [\dot{\mathbf{y}}(t_i)^T \ \ddot{\mathbf{y}}(t_i)^T]$ are computed exactly, without noise. The validation set $\mathbf{Y}_V$ is generated using the same intervals and duration, but for initial conditions: $(y_0, \dot{y}_0) \in \{(0.84, -0.11), (0.77, -0.12), (0.83, -0.13), (0.79, -0.13), (0.79, -0.10), (0.82, -0.11)\}$.

### (ii) Hybrid-SINDy discovers flight and hopping regimes

In this case, the position and velocity measurements in phase space provide a natural, data-driven coordinate system to cluster samples. Here, we identify $m = 492$ clusters, one for each timepoint. Figure 2$a$ illustrates three of these clusters. We use a model library containing polynomials up to second order in terms of $\mathbf{X}_T$. Applying SINDy to each cluster, we produce a set of models for each cluster and rank them within the cluster using relative $AIC_c$; this procedure is illustrated in figure 2$b$. We retain only the models with strong support, relative $AIC_c < 3$. Figure 2$c$ shows that the correct models are the most frequently identified by Hybrid-SINDy. In addition, when we plot the location of the discovered models in data-driven coordinates (phase space for this example), we clearly identify the compression model when $y < 1$ (teal) and the flying model when $y > 1$ (purple). There is a transition region at $y = 1$, where the incorrect models, plotted in gold and yellow, are the lowest $AIC_c$ models in the cluster.

To investigate the success of model discovery over time, figure 3 illustrates the discovered models (same colour scheme as in figure 2), the estimated model coefficients and the associated absolute $AIC_c$ values. The four switching points between compression (teal) and flying (purple) area clearly visible, with incorrect models (gold and yellow) marking each transition. The model coefficients are consistent within either the compression or flying region, but become large within the transition regions, shown in figure 3$b$.

The $AIC_c$ plot shows only the lowest absolute $AIC_c$ found in each cluster for the top four most frequent models across clusters. There is a substantial difference between the $AIC_c$ values for the correct ($AIC_c \leq 3 \times 10^{-3}$) and incorrect models ($AIC_c \geq 2 \times 10^{-2}$). As the system approaches a transition event, the $AIC_c$ for Hybrid-SINDy increases significantly. The increase is likely due to two factors: (i) as we approach the event there are fewer time points contributing to the $AIC_c$ calculation, and (ii) we find an inaccurate proposed switching point, $t_s$, for validation data. As a switching event is approached, locating $t_s$ becomes challenging, and points from after a transition are occasionally included in the local error approximation $\epsilon(k)$. Note that the increase in $AIC_c$ between clusters provides a more robust indication of the switch than Matlab's built in function *findchangepoints* applied to the time series without clustering. The *findchangepoints*, which uses statistical methods to detect change points, often fails for dynamic behaviour such as oscillations.

**Table 1.** School calendar for a year.

| session | days | time period (months) | transmission rate |
|---|---|---|---|
| winter break | 0–35 | 1.2 | $\beta = 5.2$ |
| spring term | 35–155 | 4 | $\beta = 16.8$ |
| summer break | 155–225 | 2.3 | $\beta = 5.2$ |
| fall term | 225–365 | 4.6 | $\beta = 16.8$ |

## (b) SIR disease model with switching transmission rates

In this section, we investigate a time-dependent hybrid dynamical system. Specifically, we focus on the Susceptible, Infected and Recovered (SIR) disease model with varying transmission rates. This dynamical system has been widely studied in the epidemiological community due to the nonlinear dynamics [1] and the related observations from data [91]. For example, the canonical SIR model can be modified to increase transmission rates among children when school is in session due to the increased contact rate [92]. Figure 4a illustrates the switching behaviour. The following is a description of this model:

$$\dot{S} = \nu N - \frac{\beta(t)}{N} IS - dS, \tag{4.3a}$$

$$\dot{I} = \frac{\beta(t)}{N} IS - (\gamma + d)I \tag{4.3b}$$

and

$$\dot{R} = \gamma I - dR, \tag{4.3c}$$

where $\nu = 1/365$ is the rate which students enter the population, $d = \nu$ is the rate at which students leave the population, $N = 1000$ is the total population of students, and $\gamma = 1/5$ is the recovery rate when 5 days is the average infectious period. The time-varying rate of transmission, $\beta(t)$, takes on two discrete values when school is in or out of session:

$$\beta(t) = \begin{cases} \hat{\beta}(1 + b), & t \in \text{ school in session,} \\ \hat{\beta}\dfrac{1}{(1 + b)}, & t \in \text{ school out of session.} \end{cases} \tag{4.4}$$

The variable $\hat{\beta} = 9.336$ sets a base transmission rate for students and $b = 0.8$ controls the change in transmission rate. The school year is composed of in-class sessions and breaks. The timing of these periods is outlined in table 1. We chose these slightly irregular time periods, creating a time series with annual periodicity, but no sub-annual periodicity. A lack of sub-annual periodicity could make dynamic switching hard to detect using a frequency analysis alone.

### (i) Generating input time series from the SIR model

To produce training time series, we simulate the model for 5 years, recording at a daily interval. This produces 1825 time points. We collect data along a single trajectory starting from the initial condition at $S_0 = 12$, $I_0 = 13$, $R_0 = 975$. For this model, the dynamic trajectory rapidly settles into a periodic behaviour, where the size of spring and fall outbreaks is the same for each year. We add a random perturbation to the start of each session by changing the number of children within the $S$, $I$ and $R$ state independently by either $-2$, $-1$, $0$, $1$ or $2$ children with equal probability. Over 5 years, this results in 19 perturbations, not including the initial condition. In reality, child attendance in schools will naturally fluctuate over time. These perturbations also help in identifying the correct model by perturbing the system off of the attractor. In this example, the training and validation sets rely solely on $S$ and $I$ such that $\mathbf{Y}_T = [\mathbf{S}(t_i)^T \ \mathbf{I}(t_i)^T]$. The validation time series, $\mathbf{Y}_V$, are constructed with the same number of temporal samples from a new initial condition $S_0 = 15$, $I_0 = 10$, $R_0 = 975$.

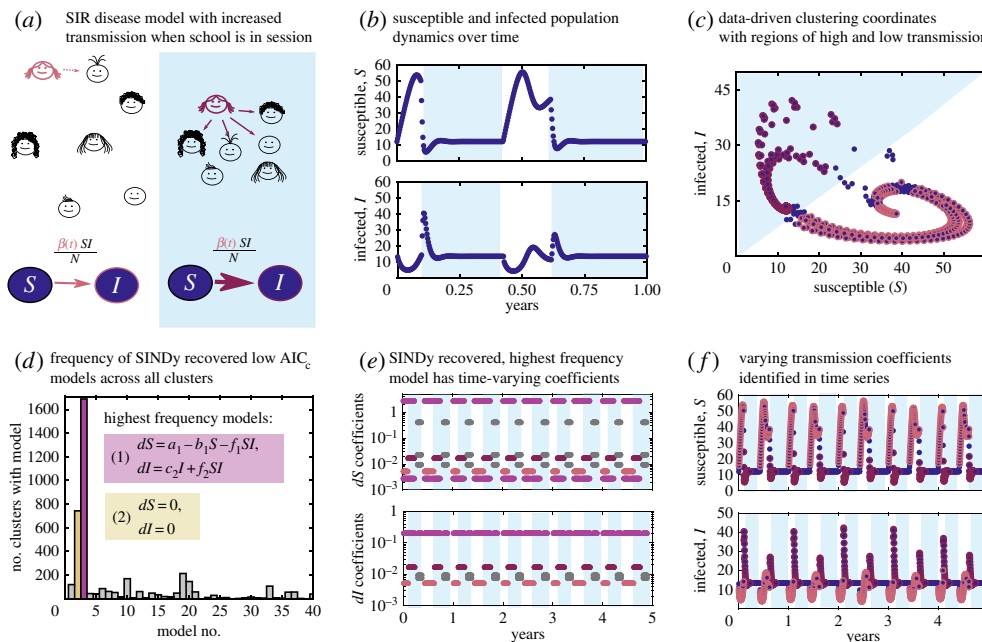

**Figure 4.** Sparse selection of Susceptible-Infected-Recovered (SIR) disease model with varying transmission rates. (*a*) School children have lower transmission rates during school breaks (white background), and higher transmission due to increased contact between children while school is in session (blue background). The infected, *I*, and susceptible, *S*, population dynamics over one school year, show declines in the infected population while school is out of session, followed by spikes or outbreaks when school is in session as shown in (*b*). Clustering in data-driven coordinates *S* versus *I*, shown in (*c*), and performing SINDy on the clusters, identifies a region with high transmission rate (maroon) and low transmission rate (pink). A frequency analysis across all clusters of the low AIC$_c$ models in each cluster, shown in (*d*), identifies two models of interest. The highest frequency model is the correct model, and SINDy has recovered the true coefficients for this model in both high and low transmission regimes. (*e*) The coefficients of the highest frequency model recovered in time. (*f*) Overlays the recovered transmission rates on the time-series data used for selection. (Online version in colour.)

## (ii) Hybrid-SINDy discovers the switching from school breaks

The relatively low transmission rate when school is out of session leads to an increase in the susceptible population. As school starts, the increase in mixing between children initiates a rapid increase in the infected population, illustrated in figure 4*a*,*b*. The training data for Hybrid-SINDy include the *S* and *I* time series illustrated in figure 4*f*. The validation time series is used to calculate the AIC$_c$ values. Here, we cluster the measurement data using the coordinates *S* and *I*, with $K = 30$ points per cluster. We use a model library containing polynomials up to third order in terms of $\mathbf{X}_T$. Two models appear with high frequency across a majority of the clusters. The highest frequency model identifies the correct dynamical terms described in equation (4.3). The other frequently identified model is a system with zero dynamics.

Examining the coefficients for the highest frequency model over time, we identify three reoccurring sets of coefficients, illustrated in figure 4*e*. The first set of recovered values correctly matches the coefficients for equations (4.3*a*,*b*) when school is out, the second set correctly recover coefficients for when school is in session, and the third are incorrect. Only the coefficient on the nonlinear transmission term, *IS*, changes value between the recovered in-school (pink) and out-of-school (maroon) transmission rates. The other coefficients (purple) are constant across the first two sets of coefficients.

The third set of coefficients (grey) are incorrect. However, during these periods of time the most frequently appearing model no longer has the lowest AIC$_c$. The second highest frequency model $\dot{S} = 0, \dot{I} = 0$ has the lowest AIC$_c$ values at those times. Additionally, the AIC$_c$ values are four

orders of magnitude larger than those calculated for the correct model with correct coefficients. Notably, the second highest frequency model is identified by Hybrid-SINDy for regions where $S$ and $I$ are not changing because the system has reached a temporary equilibrium. This model is locally accurate, but cannot predict the validation data once a new outbreak occurs, and thus has a high ($\text{AIC}_c \approx 10^{-3}$ to 1) compared to the correct model ($\text{AIC}_c \approx 10^{-6}$ to $10^{-8}$).

## (c) Robustness of Hybrid-SINDy to noise and cluster size

We examine the performance of Hybrid-SINDy when varying the cluster size and noise level. The effect of cluster-size is particularly important to understand the robustness of Hybrid-SINDy. In §4a, Hybrid-SINDy failed to recover the model during the transition events. This was primarily due to the inclusion of data from both the flying and hopping dynamic regions. In this case, the size of the regions where Hybrid-SINDy is not able to identify the correct model increases with cluster size. Alternatively, if the cluster size is too small, the SINDy regression procedure will not be able to recover the correct model from the library.

To investigate the impact of cluster size in SINDy's success, we perform a series of numerical experiments varying the cluster size and noise level. We generate a new set of training time series for the mass–spring hopping model consisting of time series from 100 random initial conditions normally distributed between $x_0 \in [1, 1.5]$ and $v_0 \in [0, 0.5]$. We divide the training set into the compression and flying subsets, avoiding the switching points. Clusters in the flying subset are constructed by picking the time-series point with maximum position value (highest flying point), and using a nearest neighbour clustering algorithm. By increasing $K$, the size of the clusters increase. A similar procedure is performed during the compression phase. Cluster sizes range from $K = 10$ to $14\,500$.

We also evaluated the recovery of correct model in these clusters by increasing measurement noise. Normally distributed noise with mean zero and $\epsilon$ from $10^{-4}$ to 10 was added to the position, $x$, and velocity, $v$, training and testing time-series data in $\mathbf{X}$. We computed the derivatives in $\dot{\mathbf{X}}$ exactly, isolating measurements noise from the challenge of computing derivatives from noisy data. For each cluster size and noise level, we generated 20 different noise realizations. SINDy is applied to each realization separately, and the fraction of successful model identifications are shown as the colour intensity in figure 5a. We did not perform a validation step, but directly checked whether the correct model was within the recovered set. With high fidelity, SINDy recovers the correct models for both the compression and flying clusters, when noise is relatively low and the cluster-size is relatively large, figure 5a. Interestingly, the cluster and noise-threshold are not the same for the compression and flying model. Recovery of the compression model varies with both noise and cluster size (the noise threshold increases for larger clusters). The cluster threshold, near $K > 50$ points, and noise threshold, near $\epsilon < 1$, for recovery of the flying model are independent. Notably, the flying model, which is simpler than the compression model, requires larger cluster size at the low noise limit.

## (d) Condition number and noise magnitude offers insight

To investigate the recovery patterns and the discrepancy between the compression and flying model, we calculate the condition number of $\Theta(\mathbf{X})$ for each cluster-size and noise magnitude, as shown in figure 5b. The range of condition numbers between the two dynamical regimes are notably different. Furthermore, the threshold for recovery (grey) does not follow the contours for the condition number. If we instead plot contours of condition number times noise magnitude, $\kappa\epsilon$, as shown in figure 5c, the contours for successful model discovery match well. The threshold, $\kappa\epsilon$, required for discovery of the compression model is much lower than that for the flying model.

The $\kappa\epsilon$ diagnostic can be related to the noise-induced error in the least squares solution—that is, the error in the solution of (2.4) with $\hat{\lambda} = 0$ and noise added to the observations $\mathbf{X}$. Because the SINDy algorithm converges to a local solution of (2.4) [93], the closeness of the initial least-squares iteration to the true solution gives some sense of when the algorithm will succeed. Let $\Xi$ denote

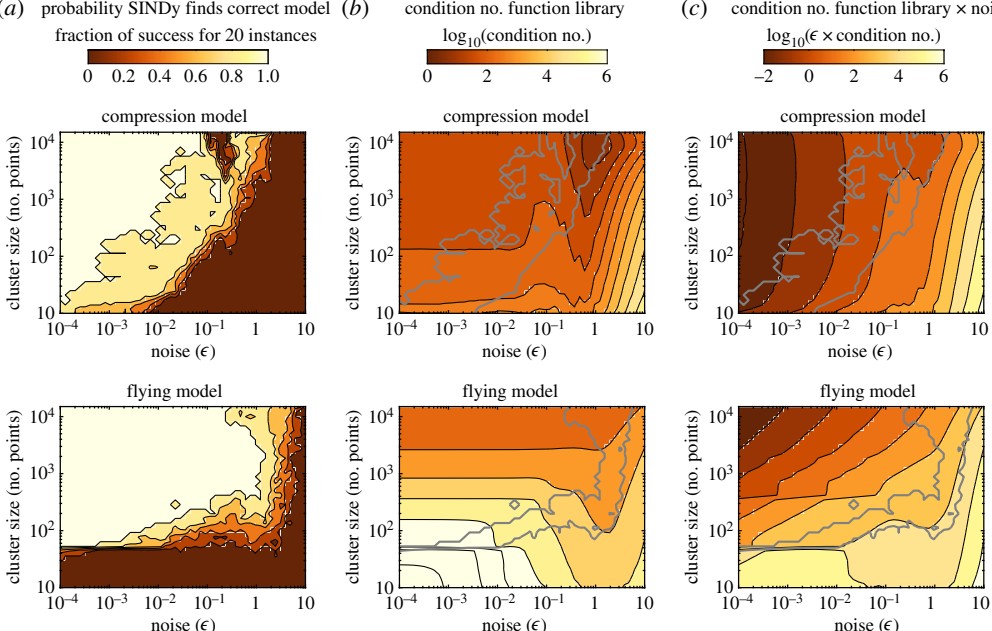

**Figure 5.** Success of Hybrid-SINDy on clustered compression (top) and flying (bottom) time-series points with varying noise (*x*-axis) and cluster size (*y*-axis). (*b*) Show the fraction of success in finding the correct model, over 20 noise-instances. When clusters are large and noise is low, models are recovered 100% of the time. The colour contours on (*b*) plots indicate $\log_{10}$ of the condition number of the function library with time series for each cluster size and noise level plugged in. (*c*) Plots show the $\log_{10}$ of the condition number of times the noise. Contours of condition number of times noise follow the contour lines for successful discovery of the model (grey). (Online version in colour.)

the true solution and $\delta \boldsymbol{\Xi}^{\mathrm{ls}}$ denote the difference between the true solution and the least-squares solution for noisy data. Then

$$\frac{\|\delta \boldsymbol{\Xi}^{\mathrm{ls}}\|_2}{\|\boldsymbol{\Xi}\|_2} \leq \frac{C\kappa\epsilon}{1 - C\kappa\epsilon}, \tag{4.5}$$

for some constant $C$ which depends only on the library functions. Note that the condition number, $\kappa$, depends on the sampling (cluster size) and choice of library functions. The complex interplay among the magnitude of noise, sampling schemes and choice of SINDy library in (4.5) provides a threshold for when we expect Hybrid-SINDy to recover the true solution. See appendix A for a more detailed discussion. The plots in figure 5*c* show that this diagnostic threshold correlates well with the empirical performance of SINDy.

It remains unclear why the particular value of $\kappa\epsilon$ for which the algorithm succeeds is three orders of magnitude higher for the flying regime than that for the compression regimes. Intuitively, there are more terms to recover and these terms have a high contrast. However, these considerations do not fully account for the difference in the observed behaviours. In appendix A, we provide some further intuition as to why the regression problem in the compression regime is more difficult than the problem in the flying regime. A more fine-grained analysis is required, taking into account the heterogeneous effect that noise in the observations has on the values of the library functions.

## 5. Discussion and conclusion

Characterizing the complex and dynamic interactions of physical and biological systems is essential for designing intervention and control strategies. For example, understanding

infectious disease transmission across human populations has led to better informed large-scale vaccination campaigns [94,95], vector control programmes [96,97] and surveillance activities [96, 98]. The increasing availability of measurement data, computational resources and data storage capacity enables new data-driven methodologies for characterization of these systems. Recent methodological innovations for identifying nonlinear dynamical systems from data have been broadly successful in a wide variety of applications including fluid dynamics [37], epidemiology [99], metabolic networks [21] and ecological systems [27,30]. The recently developed SINDy methodology identifies nonlinear models from data, offers a parsimonious and interpretable model representation [19] and generalizes well to realistic constraints such as limited and noisy data [20–22]. Broadly, SINDy is a data-analysis and modelling tool that can provide insight into mechanism as well as prediction. Despite this substantial and encouraging progress, the characterization of nonlinear systems from data is incomplete. Complex systems that exhibit switching between dynamical regimes have been far less studied with these methods, despite the ubiquity of these phenomena in physical, engineered and biological systems [2,63].

The primary contribution of this work is the generalization of SINDy to identify hybrid systems and their switching behaviour. We call this new methodology Hybrid-SINDy. By characterizing the similarity among data points, we identify clusters in measurement space using an unsupervised learning technique. A set of SINDy models is produced across clusters, and the highest frequency and most informative, predictive models are selected. We demonstrate the success of this algorithm on two modern examples of hybrid systems [2,91]: the state-dependent switching of a hopping robot and the time-dependent switching of disease transmission dynamics for children in-school and on-vacation.

For the hopping robot, Hybrid-SINDy correctly identifies the flight and compression regimes. SINDy is able to construct candidate nonlinear models from data drawn across the entire time series, but restricted to measurements similar in measurement space. This innovation allows data to be clustered based on the underlying dynamics and nonlinear geometry of trajectories, enabling the use of regression-based methods such as SINDy. The method is also quite intuitive for state-dependent hybrid systems; phase-space is effectively partitioned based on the similarity in measurement data. Moreover, this equation-free method is consistent with the underlying theory of hybrid dynamical systems by establishing charts where distinct nonlinear dynamical regimes exist between transition events. We also demonstrate that Hybrid-SINDy correctly identifies time-dependent hybrid systems from a subset of all of the phase variables. We can identify the SIR system with separate transmission rates among children during in-school versus on-vacation mixing patterns, based solely on the susceptible and infected measurements of the system. For both examples, we show that the model error characteristics and the library of candidate models help illustrate the switching behaviour even in the presence of additive measurement noise. These examples illustrate the adaptability of the method to realistic measurements and complex system behaviours.

Hybrid-SINDy incorporates the fundamental elements of a broad number of other methodologies. The method builds a library of features from measurement data to better predict the future measurement. Variations of this augmentation process have been widely explored over the last few decades, notably in the control theoretic community with delay embeddings [23–26], Carleman linearization [32] and nonlinear autoregressive models [42]. More recently, machine-learning and computer science approaches often refer to the procedure as feature engineering. Constraining the input data is another well-known approach to identify more informative and predictive models. Examples include windowing the data in time for autoregressive moving average models or identifying similarity among measurements based on the Takens' embedding theorem for delay embeddings of chaotic dynamical systems [28,30,100]. With Hybrid-SINDy, we integrate and adapt a number of these components to construct an algorithm that can identify nonlinear dynamical systems and switching between dynamical regimes.

There are limitations and challenges to the widespread adoption of our method. The method is fundamentally data-driven, requiring an adequate amount of data for each dynamical regime to perform the SINDy regression. We also rely on having access to a sufficient number

of measurement variables to construct the nonlinear dynamics, even with the inclusion of delay embeddings. These measurements also need to be in a coordinate frame to allow for a parsimonious description of the dynamics. We only consider hybrid dynamical systems without non-autonomous inputs or designed control inputs. The original SINDy procedure has been augmented to allow exogenous inputs or control [101]. To adapt Hybrid-SINDy to these systems, the clustering procedure would need to be modified to optimally cluster spatiotemporal measurements and inputs. In order to test the robustness of our results, we evaluate the condition number and noise magnitude as a numerical diagnostic for evaluating the output of Hybrid-SINDy. However, despite developing a rigorous mathematical connection between this diagnostic and numerically solving the SINDy regression, we discovered that there does not exist a specific threshold number that generalizes across models and library choice.

The k-nearest-neighbour clustering methodology was chosen as a computationally efficient, non-parametric statistical technique that does not *a priori* define statistical distributions for the data. Other clustering techniques could be easily implemented in our algorithm. For very large numbers of state-variables the clustering step may become computationally prohibitive with all techniques, potentially requiring an innovative down-sampling procedure or on-line improvement of models. Efficient on-line adaptation of models accompanied with more expensive off-line computations have been widely researched for model reduction of higher dimensional systems [102], system identification [103] and k-nearest-neighbour clustering [104]. In a future research direction, the Hybrid-SINDy methodology could be generalized to include these online–offline innovations for significantly larger systems than considered in this article.

Despite these limitations, Hybrid-SINDy is a novel step toward a general method for identifying hybrid nonlinear dynamical systems from data. We have mitigated a number of the numerical challenges by incorporating information theoretic criteria to manage uncertainty and offering a procedure to validate the results against cluster size and noise magnitude. Looking ahead, discovering a general criteria that holds across a wide variety of applications and models will be essential for the wide-spread adoption of this methodology. Furthermore, we foresee the innovative work around data-driven identification of nonlinear manifolds as another important research direction for Hybrid-SINDy [40,41].

Data accessibility. This paper contains no experimental data. All computational results are reproducible and code can be found at https://github.com/niallmm/Hybrid-SINDy.
Authors' contributions. J.L.P., N.M.M., S.L.B. and J.N.K. conceived of this work and designed the study. N.M.M. designed the algorithm and performed the computations. T.A. performed the theoretical bound analysis. N.M.M., J.L.P., T.A., S.L.B. and J.N.K. drafted the manuscript.
Competing interests. We declare we have no competing interests.
Funding. J.L.P. and N.M.M. thank Bill and Melinda Gates for their active support of the Institute for Disease Modeling and their sponsorship through the Global Good Fund. J.N.K. and T.A. acknowledge support from the Air Force Office of Scientific Research (FA9550-15-1-0385, FA9550-17-1-0329). S.L.B. and J.N.K. acknowledge support from the Defense Advanced Research Projects Agency (DARPA contract HR0011-16-C-0016). S.L.B. acknowledges support from the Army Research Office (W911NF-17-1-0422).
Acknowledgements. We are grateful to Jeff Aguilar and Daniel Goldman for discussions of their Jumping Robot system.

## Appendix A. Bound derivation

Zhang & Schaeffer [93, theorem 2.5] showed that the SINDy hard-thresholding procedure converges to a local solution of (2.4) with $R(\cdot) = \| \cdot \|_0$. Because that problem is non-convex, a local solution may or may not be equal to the true global solution. We are interested in characterizing when the initial guess for SINDy is 'close' to the exact sparse solution.

For each value of $\hat{\lambda}$, we initialize SINDy with the least-squares solution, i.e. the solution of (2.4) with $\hat{\lambda} = 0$. Noise is added to the observations $\mathbf{X}$ alone and $\dot{\mathbf{X}}$ is without noise. Let $\mathbf{X} + \delta\mathbf{X}$ denote the noisy data and let $\delta\boldsymbol{\Theta} := \boldsymbol{\Theta}(\mathbf{X} + \delta\mathbf{X}) - \boldsymbol{\Theta}(\mathbf{X})$ denote the perturbation in the resulting library.

For the sake of simplicity, we will assume that $\|\delta\boldsymbol{\Theta}\|_2/\|\boldsymbol{\Theta}\|_2 \le C\|\delta\mathbf{X}\|_2/\|\mathbf{X}\|_2 \le C\epsilon$, where $\epsilon$ is the noise level and $C$ depends only on the choice of library functions. We further assume that $\boldsymbol{\Theta}$ and $\boldsymbol{\Theta} + \delta\boldsymbol{\Theta}$ are full rank and that $\dot{\mathbf{X}} = \boldsymbol{\Theta}\,\boldsymbol{\Xi}$, i.e. that $\dot{\mathbf{X}}$ is in the range of $\boldsymbol{\Theta}$ so that the true solution $\boldsymbol{\Xi}$ satisfies $\boldsymbol{\Xi} = \boldsymbol{\Theta}^\dagger\dot{\mathbf{X}}$, where $\dagger$ denotes the Moore–Penrose pseudo-inverse.

The solution of the noisy least-squares problem is then $\boldsymbol{\Xi} + \delta\boldsymbol{\Xi}^{\mathrm{ls}} = (\boldsymbol{\Theta} + \delta\boldsymbol{\Theta})^\dagger\dot{\mathbf{X}}$, where $\delta\boldsymbol{\Xi}^{\mathrm{ls}}$ denotes the resulting error. Let $\kappa = \|\boldsymbol{\Theta}\|_2\|\boldsymbol{\Theta}^\dagger\|_2$ denote the condition number of $\boldsymbol{\Theta}$. We have the bound (4.5) from the main text

$$\frac{\|\delta\boldsymbol{\Xi}^{\mathrm{ls}}\|_2}{\|\boldsymbol{\Xi}\|_2} \le \frac{C\kappa\epsilon}{1 - C\kappa\epsilon}, \tag{A 1}$$

provided that $C\kappa\epsilon < 1$. The derivation of (A 1) is non-trivial; for a reference, see [105, theorem 5.1].

To see why the flying model is easier to recover than the compression model for a given value of $\kappa\epsilon$, we consider a single step of hard thresholding. For this derivation, we consider $\boldsymbol{\Xi}$ to be a vector; this assumption holds when $\boldsymbol{\Xi}$ is not a vector, since we typically consider solving for each column of $\boldsymbol{\Xi}$ independently in the SINDy regression. Let $c$ denote the size of the smallest non-zero coefficient in $\boldsymbol{\Xi}$, i.e. $c = \min_{i,j \text{ s.t. } \Xi_{ij} \ne 0} |\Xi_{ij}|$. A single step of hard thresholding will succeed in finding the true support of $\boldsymbol{\Xi}$ using the threshold $c/2$ when $\|\delta\boldsymbol{\Xi}^{\mathrm{ls}}\|_\infty$ is smaller than $c/2$. Let $k$ be the number of non-zero entries in $\boldsymbol{\Xi}$. Observing that $\|\boldsymbol{\Xi}\|_2 \le \sqrt{k}\|\boldsymbol{\Xi}\|_\infty$, we have

$$\frac{\|\delta\boldsymbol{\Xi}^{\mathrm{ls}}\|_\infty}{c} \le \frac{\sqrt{k}\|\boldsymbol{\Xi}\|_\infty}{c}\frac{\|\delta\boldsymbol{\Xi}^{\mathrm{ls}}\|_2}{\|\boldsymbol{\Xi}\|_2} \le \frac{\sqrt{k}\|\boldsymbol{\Xi}\|_\infty}{c}\frac{C\kappa\epsilon}{1 - C\kappa\epsilon}. \tag{A 2}$$

We see that the number of non-zero coefficients and the ratio of the largest to smallest coefficients in the true solution affect the success of a single step of hard thresholding. Intuitively, then, the compression model is more difficult to recover than the simpler flying model. However, the factor $\sqrt{k}\|\boldsymbol{\Xi}\|_\infty/c$ only accounts for about an order of magnitude of the discrepancy in the $\kappa\epsilon$ threshold at which SINDy correctly recovered compression and flying models in figure 5. A likely culprit for the remaining difference is the variation in the effect of noise on different basis functions in the library. For example, adding noise to $\mathbf{X}$ has no effect on the constant term, but will be magnified by a quadratic term. A more fine-grained analysis of the error corresponding to the specific functions in the model could account for the remaining discrepancy.

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
