## [Reviewer comments · Proceedings. Mathematical, Physical, and Engineering Sciences]

Review History

RSPA-2018-0534.R0 (Original submission)

Review form: Referee 1

Is the manuscript an original and important contribution to its field?

Yes

Is the paper of sufficient general interest?

Yes

Is the overall quality of the paper suitable?

Yes

Quality of the paper

An excellent paper making an important contribution to the field: should be published.

Can the paper be shortened without overall detriment to the main message?

Yes

Do you think some of the material would be more appropriate as an electronic appendix?

No

For papers with colour figures – is colour essential?

Yes

If there is supplementary material, is this adequate and clear?

Yes

Are there details of how to obtain materials and data, including any restrictions that may apply?

Yes

Do you have any ethical concerns with this paper?

No

Recommendation?

Accept with minor revision (please list in comments)

Comments to the Author(s)

The authors have further developed their SINDy method to hybrid systems. They clustered the collected data into multiple sub-domains and applied their SINDy method for each local data sets. One novel aspect is that to identify which model to select, they employed information theory. The proposed method is then applied to multiple interesting problems. Overall, this is very well written manuscript, and I recommend for its publication with the following minor comments to make it even stronger:

1. The authors may want to provide some comments in terms of how the optimal number of clusters can be obtained. As described in the manuscript, the cluster number is an important factor that affects the model accuracy significantly.
2. It is not clear whether the authors are considering a situation that multiple models are selected and unselected within a cluster. If it is not, would it be more beneficial to consider that?
3. In general, clustering is a computationally demanding step. I am wondering whether the authors implemented a special technique to deal with that.
4. Is the proposed method applicable on-line? Adapting/improving models on-line can be an alternative to spending a lot of efforts to collect big data. Please provide a comment on this idea.
5. Sometimes clustering may not provide a good result by putting less relevant snapshot (or, equivalently spatiotemporal data) into a same cluster. To prevent this issue, it would be beneficial to cluster them considering inputs as well as states. I am wondering whether the authors have thought about this idea.
6. There are recent contributions of developing local models and its application for model-based controller design and parameter estimation. Since they are relevant to this work, the authors may want to discuss the following papers in Introduction:

A Narasingam, P Siddhamshetty, JSI Kwon, "Handling Spatial Heterogeneity in Reservoir Parameters Using Proper Orthogonal Decomposition Based Ensemble Kalman Filter for Model-Based Feedback Control of Hydraulic Fracturing", *Industrial & Engineering Chemistry Research* 57 (11), 3977-3989

A Narasingam, JSI Kwon, "Development of local dynamic mode decomposition with control: Application to model predictive control of hydraulic fracturing", *Computers & Chemical Engineering* 106, 501-511

A Narasingam, P Siddhamshetty, JSI Kwon, "Temporal clustering for order reduction of nonlinear parabolic PDE systems with time-dependent spatial domains: Application to a hydraulic fracturing process" *AIChE Journal* 63 (9), 3818-3831

Furthermore, SINDy method has been also applied to a chemical process as follows:
A Narasingam, JSI Kwon, "Data-driven identification of interpretable reduced-order models using sparse regression", Computers & Chemical Engineering (In Press)

Review form: Referee 2

Is the manuscript an original and important contribution to its field?

Yes

Is the paper of sufficient general interest?

Yes

Is the overall quality of the paper suitable?

Yes

Quality of the paper

A good paper worth publishing in Proceedings.

Can the paper be shortened without overall detriment to the main message?

No

Do you think some of the material would be more appropriate as an electronic appendix?

No

For papers with colour figures – is colour essential?

Yes

If there is supplementary material, is this adequate and clear?

No

Are there details of how to obtain materials and data, including any restrictions that may apply?

Yes

Do you have any ethical concerns with this paper?

No

Recommendation?

Major revision is needed (please make suggestions in comments)

Comments to the Author(s)

This paper considers a generalization of the authors' previous work on sparse identification of nonlinear systems (SINDy) to the setting of hybrid systems, where the trajectory is piecewise-continuous over intervals of time, with the switching times unknown a priori, and the governing equations for each interval of time are fixed, and well-modeled by a sparse expansion in a known dictionary.

The proposed approach for identifying simultaneously the change points and the sparse dynamics within each interval is to modify the SINDy algorithm by first identifying clusters (via

k-nearest neighbors) in the measurements and identifying a set of possibly coefficients in the governing equations to describe each cluster.

This is a very important and practical problem of general interest. However, substantial changes need to be made to put the results in context of other work in the area.

First, the introduction should be re-focused. The idea of reconstructing nonlinear dynamical systems by lifting to a linear system using nonlinear function libraries was not first proposed by SINDy, even though SINDy also incorporates sparsity which is a nice contribution. Previous references for the nonlinear function library approach include

Crutchfield, McNamara. Equations of Motion from a Data Series. *Complex Systems* (1987).

Yao, Bollt. Modeling and nonlinear parameter estimation with Kronecker product representation for coupled oscillators and spatiotemporal systems. *Physica D*, 2007.

Second, some recent papers by Schaeffer, Tran, et al such as "Extracting sparse high-dimensional dynamics from limited data" and "Extracting structured dynamical systems using sparse optimization with very few samples" provide rigorous guarantees in case the nonlinear dictionary for the sparse governing equations consists of multivariate polynomials, which is indeed the setting for the examples considered in this paper. Those papers suggest that the sparse coefficients in each governing equation can be identified using a very small number of samples using a LASSO reconstruction method, thus, applying LASSO on short temporal segments of a hybrid system should work well for identifying the sparse coefficients and isolate switching times. Of course, the length of the segments in that approach is a hyperparameter that the performance will depend on, but the modifying SINDy algorithm here also has the number of clusters k as a hyperparameter, and also seems to be less rigorously motivated as clustering is hard to provide guarantees for (k -means clustering is NP hard in general). A proper discussion and numerical comparison between the approaches should be made.

Decision letter (RSPA-2018-0534.R0)

16-Nov-2018

Dear Dr Mangan

The Editor of Proceedings A has now received comments from referees on the above paper and would like you to revise it in accordance with their suggestions which can be found below (not including confidential reports to the Editor).

Please submit a copy of your revised paper within four weeks - if we do not hear from you within this time then it will be assumed that the paper has been withdrawn. In exceptional circumstances, extensions may be possible if agreed with the Editorial Office in advance.

Please note that it is the editorial policy of Proceedings A to offer authors one round of revision in which to address changes requested by referees. If the revisions are not considered satisfactory by the Editor, then the paper will be rejected, and not considered further for publication by the journal. In the event that the author chooses not to address a referee's comments, and no scientific

justification is included in their cover letter for this omission, it is at the discretion of the Editor whether to continue considering the manuscript.

- Ethics statement
- Data accessibility
- Competing interests
- Authors' contributions
- Acknowledgements
- Funding statement

See <http://royalsocietypublishing.org/instructions-authors#question3> for further details.

To revise your manuscript, log into <http://mc.manuscriptcentral.com/prsa> and enter your Author Centre, where you will find your manuscript title listed under "Manuscripts with Decisions." Under "Actions," click on "Create a Revision." Your manuscript number has been appended to denote a revision.

You will be unable to make your revisions on the originally submitted version of the manuscript. Instead, revise your manuscript and upload a new version through your Author Centre.

When submitting your revised manuscript, you will be able to respond to the comments made by the referee(s) and upload a file "Response to Referees" in "Section 6 - File Upload". Please use this to document how you have responded to the comments, and the adjustments you have made. In order to expedite the processing of the revised manuscript, please be as specific as possible in your response to the referee(s).

IMPORTANT: Your original files are available to you when you upload your revised manuscript. Please delete any unnecessary previous files before uploading your revised version.

When revising your paper please ensure that it remains under 28 pages long. In addition, any pages over 20 will be subject to a charge (£150 + VAT (where applicable) per page). Your paper has been ESTIMATED to be 21 pages.

Once again, thank you for submitting your manuscript to Proc. R. Soc. A and I look forward to receiving your revision. If you have any questions at all, please do not hesitate to get in touch.

Yours sincerely

Alice Power
Publishing Editor
Proceedings A
proceedingsa@royalsociety.org

Reviewer(s)' Comments to Author:

Referee: 1

Comments to the Author(s)

The authors have further developed their SINDy method to hybrid systems. They clustered the collected data into multiple sub-domains and applied their SINDy method for each local data sets. One novel aspect is that to identify which model to select, they employed information theory. The proposed method is then applied to multiple interesting problems. Overall, this is very well written manuscript, and I recommend for its publication with the following minor comments to make it even stronger:

1. The authors may want to provide some comments in terms of how the optimal number of clusters can be obtained. As described in the manuscript, the cluster number is an important factor that affects the model accuracy significantly.
2. It is not clear whether the authors are considering a situation that multiple models are selected and unselected within a cluster. If it is not, would it be more beneficial to consider that?
3. In general, clustering is a computationally demanding step. I am wondering whether the authors implemented a special technique to deal with that.
4. Is the proposed method applicable on-line? Adapting/improving models on-line can be an alternative to spending a lot of efforts to collect big data. Please provide a comment on this idea.
5. Sometimes clustering may not provide a good result by putting less relevant snapshot (or, equivalently spatiotemporal data) into a same cluster. To prevent this issue, it would be beneficial to cluster them considering inputs as well as states. I am wondering whether the authors have thought about this idea.
6. There are recent contributions of developing local models and its application for model-based controller design and parameter estimation. Since they are relevant to this work, the authors may want to discuss the following papers in Introduction:
 A Narasingam, P Siddhamshetty, JSI Kwon, "Handling Spatial Heterogeneity in Reservoir Parameters Using Proper Orthogonal Decomposition Based Ensemble Kalman Filter for Model-Based Feedback Control of Hydraulic Fracturing", *Industrial & Engineering Chemistry Research* 57 (11), 3977-3989
 A Narasingam, JSI Kwon, "Development of local dynamic mode decomposition with control: Application to model predictive control of hydraulic fracturing", *Computers & Chemical Engineering* 106, 501-511
 A Narasingam, P Siddhamshetty, JSI Kwon, "Temporal clustering for order reduction of nonlinear parabolic PDE systems with time-dependent spatial domains: Application to a hydraulic fracturing process" *AIChE Journal* 63 (9), 3818-3831
 Furthermore, SINDy method has been also applied to a chemical process as follows:
 A Narasingam, JSI Kwon, "Data-driven identification of interpretable reduced-order models using sparse regression", *Computers & Chemical Engineering* (In Press)

Referee: 2

Comments to the Author(s)

This paper considers a generalization of the authors' previous work on sparse identification of nonlinear systems (SINDy) to the setting of hybrid systems, where the trajectory is piecewise-continuous over intervals of time, with the switching times unknown a priori, and the governing equations for each interval of time are fixed, and well-modeled by a sparse expansion in a known dictionary.

The proposed approach for identifying simultaneously the change points and the sparse dynamics within each interval is to modify the SINDy algorithm by first identifying clusters (via k -nearest neighbors) in the measurements and identifying a set of possibly coefficients in the governing equations to describe each cluster.

This is a very important and practical problem of general interest. However, substantial changes need to be made to put the results in context of other work in the area.

First, the introduction should be re-focused. The idea of reconstructing nonlinear dynamical systems by lifting to a linear system using nonlinear function libraries was not first proposed by SINDy, even though SINDy also incorporates sparsity which is a nice contribution. Previous references for the nonlinear function library approach include

Crutchfield, McNamara. Equations of Motion from a Data Series. *Complex Systems* (1987).

Yao, Bollt. Modeling and nonlinear parameter estimation with Kronecker product representation for coupled oscillators and spatiotemporal systems. *Physica D*, 2007.

Second, some recent papers by Schaeffer, Tran, et al such as "Extracting sparse high-dimensional dynamics from limited data" and "Extracting structured dynamical systems using sparse optimization with very few samples" provide rigorous guarantees in case the nonlinear dictionary for the sparse governing equations consists of multivariate polynomials, which is indeed the setting for the examples considered in this paper. Those papers suggest that the sparse coefficients in each governing equation can be identified using a very small number of samples using a LASSO reconstruction method, thus, applying LASSO on short temporal segments of a hybrid system should work well for identifying the sparse coefficients and isolate switching times. Of course, the length of the segments in that approach is a hyperparameter that the performance will depend on, but the modifying SINDy algorithm here also has the number of clusters k as a hyperparameter, and also seems to be less rigorously motivated as clustering is hard to provide guarantees for (k -means clustering is NP hard in general). A proper discussion and numerical comparison between the approaches should be made.

RSPA-2018-0534.R1 (Revision)

Review form: Referee 1

Is the manuscript an original and important contribution to its field?

Yes

Is the paper of sufficient general interest?

Yes

Is the overall quality of the paper suitable?

Yes

Quality of the paper

An excellent paper making an important contribution to the field: should be published.

Can the paper be shortened without overall detriment to the main message?

No

Do you think some of the material would be more appropriate as an electronic appendix?

No

For papers with colour figures - is colour essential?

Yes

If there is supplementary material, is this adequate and clear?

Yes

Are there details of how to obtain materials and data, including any restrictions that may apply?

Yes

Do you have any ethical concerns with this paper?

No

Recommendation?

Accept as is

Comments to the Author(s)

The authors have addressed my comments and I recommend for its publication.

Review form: Referee 2

Is the manuscript an original and important contribution to its field?

Yes

Is the paper of sufficient general interest?

Yes

Is the overall quality of the paper suitable?

Yes

Quality of the paper

A good paper worth publishing in Proceedings.

Can the paper be shortened without overall detriment to the main message?

No

Do you think some of the material would be more appropriate as an electronic appendix?

No

For papers with colour figures - is colour essential?

Yes

If there is supplementary material, is this adequate and clear?

Yes

Are there details of how to obtain materials and data, including any restrictions that may apply?

Yes

Do you have any ethical concerns with this paper?

No

Recommendation?

Accept as is

Comments to the Author(s)

The authors have addressed my concerns and the paper is acceptable for publication.

Decision letter (RSPA-2018-0534.R1)

Dear Dr Mangan

On behalf of the Editor, I am pleased to inform you that your manuscript entitled "Model selection for hybrid dynamical systems via sparse regression" has been accepted in its final form for publication in Proceedings A.

Our Production Office will be in contact with you in due course. You can expect to receive a proof of your article soon. Please contact the office to let us know if you are likely to be away from e-mail in the near future. If you do not notify us and comments are not received within 5 days of sending the proof, we may publish the paper as it stands.

Open access

You are invited to opt for open access, our author pays publishing model. Payment of open access fees will enable your article to be made freely available via the Royal Society website as soon as it is ready for publication. For more information about open access please visit http://royalsocietypublishing.org/site/authors/open_access.xhtml. The open access fee for this journal is £1700/\$2380/€2040 per article. VAT will be charged where applicable.

Note that if you have opted for open access then payment will be required before the article is published – payment instructions will follow shortly. If you wish to opt for open access then please inform the editorial office (proceedingsa@royalsociety.org) as soon as possible.

Your article has been estimated as being 25 pages long. Our Production Office will inform you of the exact length at the proof stage.

Proceedings A levies charges for articles which exceed 20 printed pages. (based upon approximately 540 words or 2 figures per page). Articles exceeding this limit will incur page charges of £150 per page or part page, plus VAT (where applicable).

Under the terms of our licence to publish you may post the author generated postprint (ie. your accepted version not the final typeset version) of your manuscript at any time and this can be made freely available. Postprints can be deposited on a personal or institutional website, or a recognised server/repository. Please note however, that the reporting of postprints is subject to a

media embargo, and that the status the manuscript should be made clear. Upon publication of the definitive version on the publisher's site, full details and a link should be added.

You can cite the article in advance of publication using its DOI. The DOI will take the form: 10.1098/rspa.XXXX.YYYY, where XXXX and YYYY are the last 8 digits of your manuscript number (eg. if your manuscript number is RSPA-2017-1234 the DOI would be 10.1098/rspa.2017.1234).

For tips on promoting your accepted paper see our blog post:
<https://blogs.royalsociety.org/publishing/promoting-your-latest-paper-and-tracking-your-results/>

Thank you for your submission. On behalf of the Editors of the journal, we look forward to your continued contributions to the Journal.

Best wishes

Alice Power
Proceedings A Editorial Office
proceedingsa@royalsociety.org

on behalf of
Professor G. Ambika
Board Member
Proceedings A

Reviewer(s)' Comments to Author:

Referee: 1

Comments to the Author(s)
The authors have addressed my comments and I recommend for its publication.

Referee: 2

Comments to the Author(s)
The authors have addressed my concerns and the paper is acceptable for publication.